# Beyond Motif Localization: Probing Rule-Level Signals in Synthetic Genomic Grammars

**Ramu Lakshmanan**[1]     **Rafael Peres da Silva**[2]*     **Niranjan Nagarajan**[1,2,3]*

[1]Department of Computer Science, School of Computing, National University of Singapore
[2]Genome Institute of Singapore, A*STAR, Singapore
[3]Yong Loo Lin School of Medicine, National University of Singapore

## Abstract

Attribution methods are standard tools for interpreting deep learning models in regulatory genomics, but evaluations typically focus on whether motif bases receive high importance scores. We ask whether attribution maps also capture compositional rules such as motif ordering, spacing, and logical interactions. Using synthetic DNA datasets with known ground-truth grammars, we evaluate five attribution methods on localization accuracy and rule-level consistency. For the latter, we introduce the Grammar Satisfiability Score (GSS), a metric that checks whether signed attributions satisfy the Boolean logic of the generating grammar. We find that strong motif localization coexists with poor logical faithfulness for conjunctive and context-dependent grammars, and that saliency structure persists under progressive parameter randomization.

## 1 Introduction

Deep neural networks achieve strong predictive performance on DNA sequence–based tasks (Yue et al., 2023; Alipanahi et al., 2015; Kelley et al., 2016), motivating the use of attribution methods to discover learned rules by assigning per-base importance scores and localizing motifs (Talukder et al., 2021) that often resemble known cis-regulatory elements (Avsec et al., 2021; Wang et al., 2025). In practice, attribution maps are evaluated via enrichment or overlap with motif positions, i.e., whether motif bases receive higher scores than background (Koo & Ploenzke, 2021; Prakash et al., 2021; Lemanczyk et al., 2024; Krismer et al., 2022). However, regulatory control is inherently compositional: spacing, order, and context-dependent interactions between motifs can change the sign and strength of regulatory effects (Kim & Wysocka, 2023). Motif localization alone can hence imply explanations that look plausible yet are misaligned with the underlying regulatory logic (used here in a restricted, formal sense referring to explicit compositional rules defined by synthetic motif grammars).

We introduce a synthetic benchmark that makes simple regulatory logic explicit and tests whether attribution maps reflect it along two axes: **localization** (do high-magnitude scores coincide with causal bases?) and **logical consistency** (do signed attributions satisfy the generating grammar?). Across grammars where a lightweight CNN achieves near-perfect accuracy, we find that attribution maps **(i)** achieve non-trivial but limited motif localization, **(ii)** assign substantial importance to confounding patterns, **(iii)** maintain localization even as model parameters are progressively randomized, and **(iv)** misallocate signed responsibility across interacting motifs.

## 2 Methodology

### 2.1 Synthetic data and predictive model

We generated synthetic DNA classification datasets with seqgra (Krismer et al., 2022), where labels are specified by explicit motif grammars with controlled ordering and spacing constraints. We studied

---

*Corresponding authors: Rafael Peres da Silva (peres_da_silva_rafael@a-star.edu.sg) and Niranjan Nagarajan (niranjan@nus.edu.sg)

unary and pairwise rules (**NOT**, **AND_XOR**, **AND_NAND**, **XOR_XNOR**, **NIMPLY**), together with two stress tests: a frequent non-causal motif (**DUMMY**) and a grammar dependent on the presence of at least 3 motifs in any order (**COUNT-3**). For prediction, we trained a lightweight 1D CNN on one-hot encoded sequences. Across datasets, models reach 93–99% test AUROC (Appendix A, Figure 2), indicating that these grammars are learnable in this controlled setting. Full details on the dataset and model are provided in Appendix B and C.1

## 2.2 ATTRIBUTION METHODS

We evaluated Gradient×Input (Shrikumar et al., 2019), Integrated Gradients (Sundararajan et al., 2017), DeepLIFT (Shrikumar et al., 2019), Guided Grad-CAM (Selvaraju et al., 2017), and DeepSHAP (Chen et al., 2022). Each method produces base-level scores and has been examined in prior comparative studies (Prakash et al., 2021). Attributions are computed with respect to the positive-class logit. For localization metrics, we used the absolute magnitude of attribution $|a_i|$, where $a_i$ denotes the attribution score at position $i$ of the sequence. Grammar-level evaluations operate on signed attribution scores $a_i$. We denote $g_i \in \{0, 1\}$ as the presence/absence of a causal base at $i$.

## 2.3 EVALUATION

We evaluated localization of causal bases, selectivity against frequent non-causal patterns, consistency with the generating grammar, and sensitivity to model parameters as detailed below.

**Localization.** We report token-level AUPRC values (Koo & Ploenzke, 2021) that flattens all positions across sequences into a single list for all $|a_i|$ with labels $g_i$ to compute AUPRC, measuring general separation between causal motif bases and background. In addition, we propose Top-$k$ precision wherein we rank positions within each sequence by $|a_i|$, and compute the mean proportion of the top-$k$ positions that fall within causal motifs, where $k$ is the total number of motif bases to measure "purity" with. The concept is adapted from ROAR (Hooker et al., 2019), where the top-$k$ highest attribution values are removed from the data and the model is retrained as those regions are expected to be the most meaningful.

**Selectivity.** To assess whether relevance concentrates on causal motifs rather than on statistically frequent but non-causal patterns, we use enrichment-style ratios. The *Causal Relevance Score* (CRS) compares the mean attribution magnitude on causal bases to the mean magnitude on background bases. For **DUMMY**, we additionally report a *Dummy Relevance Score* (DRS), which compares the mean magnitude on dummy-motif bases to that on causal motif bases; values closer to 0 indicate greater selectivity. Formal definitions are provided in Appendix C.2.

**Grammar satisfiability.** We define the *Grammar Satisfiability Score* (GSS), a rule-level metric that evaluates the mean signed attribution maps satisfying sign and spatial predicates implied by a grammar. Given a sequence $x$ and attributions $a$, we construct predicates capturing motif presence, mean-sign constraints on motif instances, and attribution evidence in expected partner windows under grammar-specific spacing constraints. Predicates involving attribution in windows corresponding to absent motifs are exploratory and probe counterfactual behavior; violations of these clauses should not be interpreted as localization errors. Each dataset specifies a ground-truth Boolean formula over predicates. $GSS_{pos}$ and $GSS_{neg}$ are the proportion of correctly predicted sequences where the induced predicate assignment for positive and negative labels respectively evaluates as `true`. Predicate and clause definitions are provided in Appendix C.3 and Appendix D.

**Model faithfulness.** We applied cascading randomization (Adebayo et al., 2020), iteratively tracking Spearman correlation between attribution maps of the original model and those produced after sequentially randomizing the model. We compute the correlation of both signed and absolute scores.

# 3 RESULTS

## 3.1 ATTRIBUTION MAPS OFTEN LOCALIZE MOTIFS, BUT TOP-RANKED BASES CAN BE MIXED

Across grammars, attribution methods tend to assign larger-magnitude scores to motif bases than to background positions. Table 1 summarizes token-level AUPRC and within-sequence top-$k$ precision

for the best-performing configurations among the evaluated baselines (see Appendix A, Tables 12–14 for results across all baselines). In this synthetic setting, localization metrics are generally better than chance. For example, under the **AND_NAND** grammar, a random ranking yields an AUPRC of approximately $0.06$, reflecting the underlying motif prevalence, whereas all evaluated methods achieve AUPRC values above $0.6$, indicating substantially better-than-random motif localization.

At the same time, top-$k$ precision revealed that the highest-ranked bases may include a non-trivial fraction of non-causal positions for certain grammars. On **AND_XOR**, fewer than $55\%$ of the top-ranked bases fall within causal motifs, and this proportion remains at or below $0.9$ even for the unary **NOT** grammar. These results suggest caution when interpreting the magnitude of per-base attribution scores as direct indicators of importance. Instead, aggregation strategies such as motif clustering may provide a more reliable basis for localization analyses (Shrikumar et al., 2020).

| Dataset | IG | | DeepLIFT | | DeepSHAP | | Grad×Input | | Guided GradCAM | |
|---|---|---|---|---|---|---|---|---|---|---|
| | AUPRC | Top-$k$ | AUPRC | Top-$k$ | AUPRC | Top-$k$ | AUPRC | Top-$k$ | AUPRC | Top-$k$ |
| NOT | $0.88_{\pm0.02}$ | $0.87_{\pm0.02}$ | $0.88_{\pm0.01}$ | $0.90_{\pm0.02}$ | $0.88_{\pm0.01}$ | $0.90_{\pm0.02}$ | $0.59_{\pm0.04}$ | $0.82_{\pm0.02}$ | $0.01_{\pm0.00}$ | $0.01_{\pm0.01}$ |
| OR | $0.74_{\pm0.02}$ | $0.86_{\pm0.02}$ | $0.74_{\pm0.01}$ | $0.85_{\pm0.02}$ | $0.74_{\pm0.01}$ | $0.85_{\pm0.02}$ | $0.66_{\pm0.02}$ | $0.79_{\pm0.02}$ | $0.74_{\pm0.02}$ | $0.88_{\pm0.01}$ |
| NIMPLY | $0.77_{\pm0.03}$ | $0.80_{\pm0.03}$ | $0.75_{\pm0.02}$ | $0.81_{\pm0.03}$ | $0.75_{\pm0.02}$ | $0.81_{\pm0.03}$ | $0.61_{\pm0.04}$ | $0.67_{\pm0.03}$ | $0.31_{\pm0.09}$ | $0.39_{\pm0.11}$ |
| COUNT_3 | $0.88_{\pm0.02}$ | $0.83_{\pm0.02}$ | $0.90_{\pm0.02}$ | $0.85_{\pm0.02}$ | $0.90_{\pm0.02}$ | $0.85_{\pm0.02}$ | $0.76_{\pm0.03}$ | $0.78_{\pm0.02}$ | $0.79_{\pm0.01}$ | $0.80_{\pm0.03}$ |
| DUMMY | $0.73_{\pm0.02}$ | $0.61_{\pm0.02}$ | $0.76_{\pm0.02}$ | $0.62_{\pm0.02}$ | $0.76_{\pm0.02}$ | $0.62_{\pm0.02}$ | $0.62_{\pm0.02}$ | $0.56_{\pm0.02}$ | $0.62_{\pm0.01}$ | $0.49_{\pm0.03}$ |
| AND_XOR | $0.69_{\pm0.02}$ | $0.54_{\pm0.04}$ | $0.70_{\pm0.02}$ | $0.55_{\pm0.04}$ | $0.70_{\pm0.02}$ | $0.55_{\pm0.04}$ | $0.63_{\pm0.02}$ | $0.53_{\pm0.04}$ | $0.61_{\pm0.05}$ | $0.45_{\pm0.04}$ |
| XOR_XNOR | $0.75_{\pm0.03}$ | $0.82_{\pm0.03}$ | $0.77_{\pm0.01}$ | $0.86_{\pm0.01}$ | $0.77_{\pm0.01}$ | $0.86_{\pm0.01}$ | $0.68_{\pm0.03}$ | $0.81_{\pm0.02}$ | $0.21_{\pm0.07}$ | $0.30_{\pm0.12}$ |
| AND_NAND | $0.72_{\pm0.02}$ | $0.57_{\pm0.04}$ | $0.73_{\pm0.01}$ | $0.58_{\pm0.03}$ | $0.73_{\pm0.01}$ | $0.58_{\pm0.03}$ | $0.66_{\pm0.03}$ | $0.54_{\pm0.04}$ | $0.68_{\pm0.04}$ | $0.53_{\pm0.03}$ |

Table 1: Localization performance: token-level AUPRC and top-$k$ precision (Top-$k$) for each dataset and attribution method. Values are computed on the test set.

## 3.2 Non-causal motifs can frequently attract non-negligible attribution

To assess selectivity, we consider the **DUMMY** grammar, in which a high-frequency motif is embedded independently of the class label. Table 2 summarizes motif enrichment using the causal-to-background ratio (CRS) and the dummy-to-causal ratio (DRS) values. Across methods, attribution magnitudes are generally higher for causal motifs than for background positions; however, the dummy motif also receives substantial attribution, with DRS values in the range of approximately $2$ to $3$ for most methods, and only Guided Grad-CAM yielding values below $1$.

Motif occlusion experiments confirmed that the dummy motif is causally irrelevant, with a median change in the output logit of approximately $0.04$ upon occlusion (See Appendix A Table 8). Taken together, these results are consistent with previous findings that saliency-based explanations may emphasize statistically frequent or regular sequence patterns, even when such patterns are weakly coupled to the model output (Adebayo et al., 2020).

| Dataset | IG | | DeepLIFT | | DeepSHAP | | Grad×Input | | Guided GradCAM | |
|---|---|---|---|---|---|---|---|---|---|---|
| | CRS | DRS | CRS | DRS | CRS | DRS | CRS | DRS | CRS | DRS |
| DUMMY | $8.94_{\pm0.46}$ | $1.96_{\pm1.11}$ | $9.40_{\pm0.39}$ | $2.97_{\pm2.00}$ | $9.40_{\pm0.39}$ | $2.97_{\pm2.00}$ | $8.82_{\pm0.47}$ | $2.55_{\pm1.53}$ | $8.30_{\pm1.45}$ | $0.52_{\pm0.08}$ |

Table 2: Enrichment results on the **DUMMY** dataset. CRS: Causal Relevance Score (causal vs background). DRS: Dummy Relevance Score (dummy vs causal).

## 3.3 Cascading randomization reveals limited sensitivity to compositional structure

As a faithfulness check, we applied cascading randomization and tracked Spearman correlation between attribution maps before and after progressively reinitializing model layers (Adebayo et al., 2020). We consistently found that correlations computed on signed attributions decay earlier than those computed on absolute values.

Concretely, after both linear layers are randomized, absolute attributions can retain moderately high correlation with the original maps, whereas signed correlations drop markedly. Absolute correlations only collapse once early convolutional layers are randomized. This pattern suggests that the spatial support for the locations of high-magnitude regions is driven largely by motif detection encoded in the convolutional feature extractor, while the sign structure is more dependent on later layers.

Figure 1 summarizes the signed and absolute-value correlation results. DeepSHAP was omitted from cascading randomization due to resource constraints.

Figure 1: Cascading Randomization Plots for Spearman correlation

## 3.4 RULE-LEVEL CONSISTENCY VARIES ACROSS GRAMMARS

Table 3 reports the rule-level consistency score (GSS) for the best-performing attribution baselines (full results are provided in Appendix A, Tables 15–17). Disjunctive grammars, such as **OR**, generally achieve high GSS values, whereas conjunctive and context-dependent grammars show substantially lower scores despite high predictive accuracy.

The strongest deviations occur for **XOR_XNOR**, where several methods yield near-zero $GSS_{pos}$, while Grad×Input reaches $0.11$ and attains substantially higher $GSS_{neg}$ than baseline-based approaches. Overall, these results suggest that strong motif localization does not necessarily imply alignment between signed attributions and the underlying interaction rules. Details of the attribution-derived predicates used for GSS evaluation are provided in Appendix D.

| Dataset | IG | | DeepLIFT | | DeepSHAP | | Grad×Input | | Guided GradCAM | |
|---|---|---|---|---|---|---|---|---|---|---|
| | $GSS_{pos}$ | $GSS_{neg}$ | $GSS_{pos}$ | $GSS_{neg}$ | $GSS_{pos}$ | $GSS_{neg}$ | $GSS_{pos}$ | $GSS_{neg}$ | $GSS_{pos}$ | $GSS_{neg}$ |
| NOT | $1.00_{\pm 0.00}$ | $1.00_{\pm 0.00}$ | $1.00_{\pm 0.00}$ | $1.00_{\pm 0.00}$ | $1.00_{\pm 0.00}$ | $1.00_{\pm 0.00}$ | $1.00_{\pm 0.00}$ | $0.18_{\pm 0.14}$ | $1.00_{\pm 0.00}$ | $0.00_{\pm 0.00}$ |
| OR | $0.99_{\pm 0.00}$ | $1.00_{\pm 0.00}$ | $0.99_{\pm 0.00}$ | $1.00_{\pm 0.00}$ | $0.99_{\pm 0.00}$ | $1.00_{\pm 0.00}$ | $0.99_{\pm 0.00}$ | $1.00_{\pm 0.00}$ | $0.98_{\pm 0.00}$ | $1.00_{\pm 0.00}$ |
| NIMPLY | $0.27_{\pm 0.08}$ | $0.92_{\pm 0.04}$ | $0.31_{\pm 0.09}$ | $0.89_{\pm 0.06}$ | $0.31_{\pm 0.09}$ | $0.88_{\pm 0.06}$ | $0.34_{\pm 0.10}$ | $0.72_{\pm 0.09}$ | $0.64_{\pm 0.20}$ | $0.34_{\pm 0.00}$ |
| COUNT_3 | $0.94_{\pm 0.00}$ | $0.91_{\pm 0.00}$ | $0.94_{\pm 0.00}$ | $0.91_{\pm 0.00}$ | $0.94_{\pm 0.00}$ | $0.91_{\pm 0.00}$ | $0.94_{\pm 0.00}$ | $0.91_{\pm 0.00}$ | $0.00_{\pm 0.00}$ | $0.00_{\pm 0.00}$ |
| AND_XOR | $1.00_{\pm 0.00}$ | $0.10_{\pm 0.03}$ | $1.00_{\pm 0.00}$ | $0.08_{\pm 0.03}$ | $1.00_{\pm 0.00}$ | $0.08_{\pm 0.03}$ | $1.00_{\pm 0.00}$ | $0.18_{\pm 0.05}$ | $0.91_{\pm 0.09}$ | $0.00_{\pm 0.00}$ |
| DUMMY | $0.94_{\pm 0.00}$ | $0.22_{\pm 0.07}$ | $0.94_{\pm 0.00}$ | $0.17_{\pm 0.02}$ | $0.94_{\pm 0.00}$ | $0.17_{\pm 0.02}$ | $0.94_{\pm 0.00}$ | $0.22_{\pm 0.03}$ | $0.82_{\pm 0.06}$ | $0.35_{\pm 0.00}$ |
| AND_NAND | $1.00_{\pm 0.00}$ | $0.41_{\pm 0.03}$ | $1.00_{\pm 0.00}$ | $0.39_{\pm 0.02}$ | $1.00_{\pm 0.00}$ | $0.39_{\pm 0.02}$ | $1.00_{\pm 0.00}$ | $0.44_{\pm 0.03}$ | $0.91_{\pm 0.05}$ | $0.34_{\pm 0.00}$ |
| XOR_XNOR | $0.02_{\pm 0.02}$ | $0.50_{\pm 0.00}$ | $0.00_{\pm 0.00}$ | $0.50_{\pm 0.00}$ | $0.00_{\pm 0.00}$ | $0.50_{\pm 0.00}$ | $0.11_{\pm 0.01}$ | $0.96_{\pm 0.01}$ | $0.00_{\pm 0.00}$ | $0.50_{\pm 0.00}$ |

Table 3: Grammar Satisfiability Score (GSS) for positive ($GSS_{pos}$) and negative ($GSS_{neg}$) test sequences across datasets and methods.

## 4 DISCUSSION AND FUTURE WORK

This work highlights a gap between motif-level localization and evidence for compositional reasoning in genomic attribution maps. Although attribution methods often achieve strong localization in our benchmark, evaluations of logical faithfulness and sanity checks reveal discrepancies across grammars. In particular, grammar satisfiability can remain low for interaction-heavy grammars despite strong predictive accuracy, and the spatial support of saliency maps can persist under parameter randomization. Cascading randomization provides the clearest signal. While signed attributions change after parameter perturbations, the locations of high-magnitude regions can remain correlated with the original maps, with correlations decreasing mainly after early convolutional layers are randomized. This pattern is consistent with explanations being driven by motif detection in the feature extractor rather than by representations of interaction rules. We emphasize that GSS is

exploratory and captures only a limited, operational notion of consistency, as attribution methods are additive and local and do not represent counterfactual absence. Nonetheless, GSS reveals a practical limitation: in conjunctive grammars, motifs that are necessary but individually insufficient can receive negative attributions, leading to explanations that are difficult to reconcile with the underlying generative logic.

Several limitations qualify these findings. The study focuses on a single CNN architecture, and perturbation-based explanation pipelines (Novakovsky et al., 2023; Avsec et al., 2021) are not evaluated, as perturbation raises unique questions around distribution shift under unrealistic edits and combinatorial cost that warrants distinct evaluation protocols. In addition, the grammars are simple, deterministic, and synthetic. This was a deliberate choice to evaluate the interpretation of attribution maps for compositional evidence. If the failure modes raised exist in restricted cases, increasing complexity would further compound these limitations. Future directions to quantify the extent to which these results apply in realistic datasets include building sequences based on well-characterized promoter datasets (de Boer et al., 2020), as well as running cascading randomization tests on existing regulatory genomic models (de Almeida et al., 2022). Overall, our results highlight the need for evaluation protocols that directly probe compositional structure. Criteria such as GSS can help distinguish explanations that merely identify motifs from those that reflect interaction rules. We view this work as a step toward grammar-level interpretability in genomics and a foundation for evaluating future explanation methods that search over combinatorial edits to infer regulatory grammar for more realistic regulatory datasets and architectures. The code for this work is available from github at: https://github.com/R-Laksh/genomic-attr-bench.

## MEANINGFULNESS STATEMENT

Understanding life requires methodologies that capture not only which biological elements are present, but how they interact. This work contributes by examining whether common attribution methods recover compositional regulatory logic, such as motif interactions and contextual rules, rather than only local sequence features. By introducing controlled synthetic grammars and rule-level evaluation criteria, we provide tools to distinguish superficial motif localization from explanations that reflect underlying biological mechanisms. This helps clarify the limits of current interpretability methods and supports the development of models and explanations that better align with the complex combinatorial nature of gene regulation.

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

# A  APPENDIX A

## A.1  SUPPLEMENTARY FIGURES

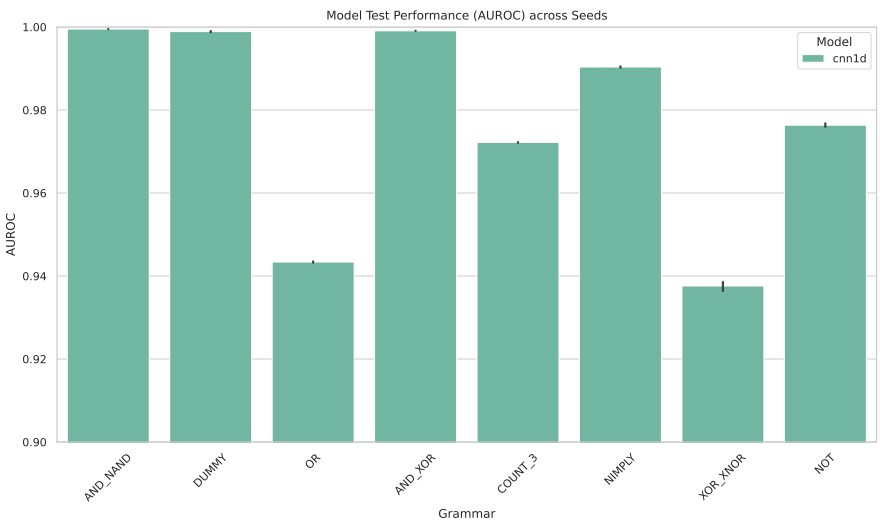

Figure 2: Test AUROC across datasets

## A.2  SUPPLEMENTARY TABLES

| Occlusion | Median $\Delta$logit | Mean $\Delta$logit |
|---|---|---|
| TATAAA | $-13.64_{\pm 1.63}$ | $-9.54_{\pm 0.76}$ |
| CCAAT | $-16.45_{\pm 1.19}$ | $-10.51_{\pm 0.72}$ |

Table 4: Results for occluding present motifs in **AND_XOR**

| Occlusion | Median $\Delta$logit | Mean $\Delta$logit |
|---|---|---|
| Spacing break | $-17.81_{\pm 1.21}$ | $-17.67_{\pm 1.19}$ |
| Order swap | $-17.78_{\pm 1.21}$ | $-17.61_{\pm 1.19}$ |

Table 5: Results for occluding motif order in **AND_XOR**

| Occlusion | Median $\Delta$logit | Mean $\Delta$logit |
|---|---|---|
| TATAAA | $-15.10_{\pm 1.38}$ | $-10.50_{\pm 0.86}$ |
| CCAAT | $-16.39_{\pm 1.23}$ | $-11.31_{\pm 0.79}$ |

Table 6: Results for occluding present motifs in **AND_NAND**

| Occlusion | Median $\Delta$logit | Mean $\Delta$logit |
|---|---|---|
| Spacing break | $-17.41_{\pm 1.29}$ | $-17.27_{\pm 1.27}$ |
| Order swap | $-17.40_{\pm 1.30}$ | $-17.28_{\pm 1.26}$ |

Table 7: Results for occluding motif order in **AND_NAND**

| Occlusion | Median $\Delta$logit | Mean $\Delta$logit |
|---|---|---|
| TATAAA | $-8.76_{\pm 0.59}$ | $-7.53_{\pm 0.49}$ |
| CCAAT | $-12.72_{\pm 0.73}$ | $-9.04_{\pm 0.56}$ |
| CGCCAT (dummy) | $0.04_{\pm 0.03}$ | $-0.52_{\pm 0.10}$ |

Table 8: Results for occlusion of motifs in **DUMMY**

| Occlusion | Median $\Delta$logit | Mean $\Delta$logit |
|---|---|---|
| Spacing break | $-14.45_{\pm 0.96}$ | $-14.02_{\pm 0.90}$ |
| Order swap | $-14.42_{\pm 0.96}$ | $-13.86_{\pm 0.92}$ |

Table 9: Results for occluding motif order in **DUMMY**

| Occlusion | Median $\Delta$logit | Mean $\Delta$logit |
|---|---|---|
| TATAAA | $0.77_{\pm 0.16}$ | $-3.14_{\pm 0.31}$ |
| CCAAT | $0.03_{\pm 0.02}$ | $-2.77_{\pm 0.30}$ |

Table 10: Results for occluding present motifs in **XOR_XNOR**

| Occlusion | Median $\Delta$logit | Mean $\Delta$logit |
|---|---|---|
| Spacing break | $-0.71_{\pm 0.28}$ | $-0.88_{\pm 0.18}$ |
| Order swap | $0.21_{\pm 0.12}$ | $0.15_{\pm 0.17}$ |

Table 11: Results for occluding motif order in **XOR_XNOR**

| Dataset | IG | | DeepLIFT | | DeepSHAP | |
|---|---|---|---|---|---|---|
| | AUPRC | Top-$k$ | AUPRC | Top-$k$ | AUPRC | Top-$k$ |
| NOT | $0.87_{\pm 0.03}$ | $0.89_{\pm 0.02}$ | $0.88_{\pm 0.01}$ | $0.90_{\pm 0.02}$ | $0.88_{\pm 0.01}$ | $0.90_{\pm 0.02}$ |
| OR | $0.73_{\pm 0.01}$ | $0.84_{\pm 0.01}$ | $0.74_{\pm 0.01}$ | $0.85_{\pm 0.02}$ | $0.74_{\pm 0.01}$ | $0.85_{\pm 0.02}$ |
| NIMPLY | $0.76_{\pm 0.03}$ | $0.79_{\pm 0.03}$ | $0.75_{\pm 0.02}$ | $0.81_{\pm 0.03}$ | $0.75_{\pm 0.02}$ | $0.81_{\pm 0.03}$ |
| COUNT_3 | $0.88_{\pm 0.02}$ | $0.83_{\pm 0.02}$ | $0.90_{\pm 0.02}$ | $0.85_{\pm 0.02}$ | $0.90_{\pm 0.02}$ | $0.85_{\pm 0.02}$ |
| DUMMY | $0.73_{\pm 0.02}$ | $0.61_{\pm 0.02}$ | $0.76_{\pm 0.02}$ | $0.62_{\pm 0.02}$ | $0.76_{\pm 0.02}$ | $0.62_{\pm 0.02}$ |
| AND_XOR | $0.69_{\pm 0.02}$ | $0.54_{\pm 0.04}$ | $0.70_{\pm 0.02}$ | $0.55_{\pm 0.04}$ | $0.70_{\pm 0.02}$ | $0.55_{\pm 0.04}$ |
| XOR_XNOR | $0.02_{\pm 0.02}$ | $0.50_{\pm 0.00}$ | $0.00_{\pm 0.00}$ | $0.50_{\pm 0.00}$ | $0.00_{\pm 0.00}$ | $0.50_{\pm 0.00}$ |
| AND_NAND | $0.72_{\pm 0.02}$ | $0.57_{\pm 0.04}$ | $0.73_{\pm 0.01}$ | $0.58_{\pm 0.03}$ | $0.73_{\pm 0.01}$ | $0.58_{\pm 0.03}$ |

Table 12: Localization performance: Zero Baseline

| Dataset | IG | | DeepLIFT | | DeepSHAP | |
|---|---|---|---|---|---|---|
| | AUPRC | Top-$k$ | AUPRC | Top-$k$ | AUPRC | Top-$k$ |
| NOT | $0.88_{\pm 0.02}$ | $0.87_{\pm 0.02}$ | $0.81_{\pm 0.04}$ | $0.83_{\pm 0.03}$ | $0.81_{\pm 0.04}$ | $0.83_{\pm 0.03}$ |
| OR | $0.74_{\pm 0.02}$ | $0.86_{\pm 0.02}$ | $0.73_{\pm 0.02}$ | $0.84_{\pm 0.02}$ | $0.73_{\pm 0.02}$ | $0.84_{\pm 0.02}$ |
| NIMPLY | $0.77_{\pm 0.03}$ | $0.80_{\pm 0.03}$ | $0.71_{\pm 0.03}$ | $0.76_{\pm 0.03}$ | $0.71_{\pm 0.03}$ | $0.76_{\pm 0.03}$ |
| COUNT_3 | $0.86_{\pm 0.02}$ | $0.82_{\pm 0.03}$ | $0.75_{\pm 0.03}$ | $0.73_{\pm 0.03}$ | $0.75_{\pm 0.03}$ | $0.73_{\pm 0.03}$ |
| DUMMY | $0.71_{\pm 0.03}$ | $0.59_{\pm 0.03}$ | $0.65_{\pm 0.04}$ | $0.55_{\pm 0.03}$ | $0.65_{\pm 0.04}$ | $0.55_{\pm 0.03}$ |
| AND_XOR | $0.65_{\pm 0.02}$ | $0.49_{\pm 0.03}$ | $0.59_{\pm 0.03}$ | $0.46_{\pm 0.03}$ | $0.59_{\pm 0.03}$ | $0.46_{\pm 0.03}$ |
| XOR_XNOR | $0.75_{\pm 0.02}$ | $0.82_{\pm 0.02}$ | $0.72_{\pm 0.03}$ | $0.81_{\pm 0.02}$ | $0.72_{\pm 0.03}$ | $0.81_{\pm 0.02}$ |
| AND_NAND | $0.69_{\pm 0.02}$ | $0.52_{\pm 0.03}$ | $0.60_{\pm 0.02}$ | $0.47_{\pm 0.03}$ | $0.60_{\pm 0.02}$ | $0.47_{\pm 0.03}$ |

Table 13: Localization performance: Uniform Baseline

| Dataset | IG | | DeepLIFT | | DeepSHAP | |
|---|---|---|---|---|---|---|
| | AUPRC | Top-$k$ | AUPRC | Top-$k$ | AUPRC | Top-$k$ |
| NOT | $0.87_{\pm 0.03}$ | $0.89_{\pm 0.02}$ | $0.88_{\pm 0.01}$ | $0.90_{\pm 0.02}$ | $0.88_{\pm 0.01}$ | $0.90_{\pm 0.02}$ |
| OR | $0.73_{\pm 0.01}$ | $0.84_{\pm 0.01}$ | $0.74_{\pm 0.01}$ | $0.85_{\pm 0.02}$ | $0.74_{\pm 0.01}$ | $0.85_{\pm 0.02}$ |
| NIMPLY | $0.76_{\pm 0.03}$ | $0.79_{\pm 0.03}$ | $0.75_{\pm 0.02}$ | $0.81_{\pm 0.03}$ | $0.75_{\pm 0.02}$ | $0.81_{\pm 0.03}$ |
| COUNT_3 | $0.88_{\pm 0.02}$ | $0.83_{\pm 0.02}$ | $0.90_{\pm 0.02}$ | $0.85_{\pm 0.02}$ | $0.90_{\pm 0.02}$ | $0.85_{\pm 0.02}$ |
| DUMMY | $0.73_{\pm 0.02}$ | $0.61_{\pm 0.02}$ | $0.76_{\pm 0.02}$ | $0.62_{\pm 0.02}$ | $0.76_{\pm 0.02}$ | $0.62_{\pm 0.02}$ |
| AND_XOR | $0.69_{\pm 0.02}$ | $0.54_{\pm 0.04}$ | $0.70_{\pm 0.02}$ | $0.55_{\pm 0.04}$ | $0.70_{\pm 0.02}$ | $0.55_{\pm 0.04}$ |
| XOR_XNOR | $0.75_{\pm 0.03}$ | $0.82_{\pm 0.03}$ | $0.77_{\pm 0.01}$ | $0.86_{\pm 0.01}$ | $0.77_{\pm 0.01}$ | $0.86_{\pm 0.01}$ |
| AND_NAND | $0.72_{\pm 0.02}$ | $0.57_{\pm 0.04}$ | $0.73_{\pm 0.01}$ | $0.58_{\pm 0.03}$ | $0.73_{\pm 0.01}$ | $0.58_{\pm 0.03}$ |

Table 14: Localization performance: Dinucleotide Shuffled Baseline

| Dataset | IG | | DeepLIFT | | DeepSHAP | |
|---|---|---|---|---|---|---|
| | $GSS_{pos}$ | $GSS_{neg}$ | $GSS_{pos}$ | $GSS_{neg}$ | $GSS_{pos}$ | $GSS_{neg}$ |
| NOT | $1.00_{\pm 0.00}$ | $1.00_{\pm 0.00}$ | $1.00_{\pm 0.00}$ | $1.00_{\pm 0.00}$ | $1.00_{\pm 0.00}$ | $1.00_{\pm 0.00}$ |
| OR | $0.99_{\pm 0.00}$ | $1.00_{\pm 0.00}$ | $0.99_{\pm 0.00}$ | $1.00_{\pm 0.00}$ | $0.99_{\pm 0.00}$ | $1.00_{\pm 0.00}$ |
| NIMPLY | $0.33_{\pm 0.10}$ | $0.88_{\pm 0.08}$ | $0.31_{\pm 0.09}$ | $0.89_{\pm 0.06}$ | $0.31_{\pm 0.09}$ | $0.88_{\pm 0.06}$ |
| COUNT_3 | $0.94_{\pm 0.00}$ | $0.91_{\pm 0.00}$ | $0.94_{\pm 0.00}$ | $0.91_{\pm 0.00}$ | $0.94_{\pm 0.00}$ | $0.91_{\pm 0.00}$ |
| DUMMY | $0.94_{\pm 0.00}$ | $0.22_{\pm 0.07}$ | $0.94_{\pm 0.00}$ | $0.17_{\pm 0.02}$ | $0.94_{\pm 0.00}$ | $0.17_{\pm 0.02}$ |
| AND_XOR | $1.00_{\pm 0.00}$ | $0.10_{\pm 0.03}$ | $1.00_{\pm 0.00}$ | $0.08_{\pm 0.03}$ | $1.00_{\pm 0.00}$ | $0.08_{\pm 0.03}$ |
| XOR_XNOR | $0.02_{\pm 0.02}$ | $0.50_{\pm 0.00}$ | $0.00_{\pm 0.00}$ | $0.50_{\pm 0.00}$ | $0.00_{\pm 0.00}$ | $0.50_{\pm 0.00}$ |
| AND_NAND | $1.00_{\pm 0.00}$ | $0.41_{\pm 0.03}$ | $1.00_{\pm 0.00}$ | $0.39_{\pm 0.02}$ | $1.00_{\pm 0.00}$ | $0.39_{\pm 0.02}$ |

Table 15: Grammar Satisfiability Score: Zero Baseline

| Dataset | IG | | DeepLIFT | | DeepSHAP | |
|---|---|---|---|---|---|---|
| | $GSS_{pos}$ | $GSS_{neg}$ | $GSS_{pos}$ | $GSS_{neg}$ | $GSS_{pos}$ | $GSS_{neg}$ |
| NOT | $1.00_{\pm 0.00}$ | $1.00_{\pm 0.00}$ | $1.00_{\pm 0.00}$ | $1.00_{\pm 0.00}$ | $1.00_{\pm 0.00}$ | $1.00_{\pm 0.00}$ |
| OR | $0.99_{\pm 0.00}$ | $1.00_{\pm 0.00}$ | $0.99_{\pm 0.00}$ | $1.00_{\pm 0.00}$ | $0.99_{\pm 0.00}$ | $1.00_{\pm 0.00}$ |
| NIMPLY | $0.27_{\pm 0.08}$ | $0.92_{\pm 0.04}$ | $0.19_{\pm 0.10}$ | $0.89_{\pm 0.05}$ | $0.19_{\pm 0.10}$ | $0.89_{\pm 0.05}$ |
| COUNT_3 | $0.94_{\pm 0.00}$ | $0.91_{\pm 0.00}$ | $0.94_{\pm 0.00}$ | $0.91_{\pm 0.00}$ | $0.94_{\pm 0.00}$ | $0.91_{\pm 0.00}$ |
| DUMMY | $0.94_{\pm 0.00}$ | $0.20_{\pm 0.04}$ | $0.94_{\pm 0.00}$ | $0.19_{\pm 0.03}$ | $0.94_{\pm 0.00}$ | $0.19_{\pm 0.03}$ |
| AND_XOR | $1.00_{\pm 0.00}$ | $0.08_{\pm 0.02}$ | $1.00_{\pm 0.00}$ | $0.09_{\pm 0.03}$ | $1.00_{\pm 0.00}$ | $0.09_{\pm 0.03}$ |
| XOR_XNOR | $0.02_{\pm 0.01}$ | $0.50_{\pm 0.00}$ | $0.00_{\pm 0.00}$ | $0.50_{\pm 0.00}$ | $0.00_{\pm 0.00}$ | $0.50_{\pm 0.00}$ |
| AND_NAND | $1.00_{\pm 0.00}$ | $0.41_{\pm 0.03}$ | $1.00_{\pm 0.00}$ | $0.41_{\pm 0.03}$ | $1.00_{\pm 0.00}$ | $0.41_{\pm 0.03}$ |

Table 16: Grammar Satisfiability Score: Uniform Baseline

| Dataset | IG | | DeepLIFT | | DeepSHAP | |
|---|---|---|---|---|---|---|
| | $GSS_{pos}$ | $GSS_{neg}$ | $GSS_{pos}$ | $GSS_{neg}$ | $GSS_{pos}$ | $GSS_{neg}$ |
| NOT | $1.00_{\pm 0.00}$ | $1.00_{\pm 0.00}$ | $1.00_{\pm 0.00}$ | $1.00_{\pm 0.00}$ | $1.00_{\pm 0.00}$ | $1.00_{\pm 0.00}$ |
| OR | $0.99_{\pm 0.00}$ | $1.00_{\pm 0.00}$ | $0.99_{\pm 0.00}$ | $1.00_{\pm 0.00}$ | $0.99_{\pm 0.00}$ | $1.00_{\pm 0.00}$ |
| NIMPLY | $0.33_{\pm 0.10}$ | $0.88_{\pm 0.08}$ | $0.31_{\pm 0.09}$ | $0.89_{\pm 0.06}$ | $0.31_{\pm 0.09}$ | $0.88_{\pm 0.06}$ |
| COUNT_3 | $0.94_{\pm 0.00}$ | $0.91_{\pm 0.00}$ | $0.94_{\pm 0.00}$ | $0.91_{\pm 0.00}$ | $0.94_{\pm 0.00}$ | $0.91_{\pm 0.00}$ |
| DUMMY | $0.94_{\pm 0.00}$ | $0.22_{\pm 0.07}$ | $0.94_{\pm 0.00}$ | $0.17_{\pm 0.02}$ | $0.94_{\pm 0.00}$ | $0.17_{\pm 0.02}$ |
| AND_XOR | $1.00_{\pm 0.00}$ | $0.10_{\pm 0.03}$ | $1.00_{\pm 0.00}$ | $0.08_{\pm 0.03}$ | $1.00_{\pm 0.00}$ | $0.08_{\pm 0.03}$ |
| XOR_XNOR | $0.02_{\pm 0.02}$ | $0.50_{\pm 0.00}$ | $0.00_{\pm 0.00}$ | $0.50_{\pm 0.00}$ | $0.00_{\pm 0.00}$ | $0.50_{\pm 0.00}$ |
| AND_NAND | $1.00_{\pm 0.00}$ | $0.41_{\pm 0.03}$ | $1.00_{\pm 0.00}$ | $0.39_{\pm 0.02}$ | $1.00_{\pm 0.00}$ | $0.39_{\pm 0.02}$ |

Table 17: Grammar Satisfiability Score: Dinucleotide Shuffled Baseline

# B APPENDIX B

All datasets use 200 bp sequences over the alphabet $\{A, C, G, T\}$ with motif occurrences embedded into background sequence sampled from a fixed nucleotide distribution. We use three short motifs: TATAAA (MOTIF-A), CCAAT (MOTIF-B), and a dummy motif CGCCAT (MOTIF-D). The grammars are:

- **NOT**: Unary grammar using MOTIF-A, where positives lack any occurrence of MOTIF-A and negatives contain at least one occurrence.
- **AND_XOR**: Binary grammar over MOTIF-A and MOTIF-B, where positives contain both motifs with MOTIF-A located 3–5 bp upstream of MOTIF-B, and negatives contain any other configuration with at least 1 motif present, including isolated motifs.

- **AND_NAND**: Binary grammar over MOTIF-A and MOTIF-B, where positives contain both motifs with MOTIF-A located 3–5 bp upstream of MOTIF-B, and negatives contain any other configuration, including isolated motifs or no motifs.

- **XOR_XNOR**: Binary grammar where positives contain exactly one of MOTIF-A or MOTIF-B and negatives contain both or neither, again enforcing a 3–5 bp spacing window when both motifs are present.

- **NIMPLY**: Binary grammar where positives contain MOTIF-A but not MOTIF-B 3–5 bp downstream, and negatives contain any other configuration, including isolated motifs.

- **DUMMY**: Based on **AND_NAND**, but with the dummy MOTIF-D inserted independently with probability 0.6 in both positive and negative examples. This grammar isolates the ability of explanations to distinguish causally relevant motifs from statistically frequent yet irrelevant patterns.

- **COUNT-3**: A counting grammar where positives contain 3, 4 or 5 non-overlapping occurrences of MOTIF-A at any position, and negatives contain fewer occurrences.

Each dataset contains 160,000 training samples, 20,000 validation samples, and 40,000 test samples, all drawn from the same underlying data distribution with balanced positive and negative classes.

# C  APPENDIX C

## C.1  MODEL ARCHITECTURE

The architecture comprises three convolutional blocks (ReLU + max pooling) followed by two fully connected layers, trained with binary cross-entropy for 4 epochs using Adam optimizer with results averaged over 10 seeds.

## C.2  SELECTIVITY METRICS

Let $a_i \in \mathbb{R}$ denote the signed attribution at position $i$ and let $g_i \in \{0, 1\}$ indicate whether position $i$ is part of a causal motif instance. We report the following magnitude-based ratios:

$$\text{CRS} = \frac{\mathbb{E}[|a_i| \mid g_i = 1]}{\mathbb{E}[|a_i| \mid g_i = 0] + \varepsilon},$$

where $\varepsilon$ is a small constant to avoid division by zero. CRS quantifies how strongly relevance concentrates on causal bases relative to background.

In the **DUMMY** dataset, let $d_i \in \{0, 1\}$ indicate dummy-motif bases. We define

$$\text{DRS} = \frac{\mathbb{E}[|a_i| \mid d_i = 1]}{\mathbb{E}[|a_i| \mid g_i = 1] + \varepsilon},$$

so that smaller values indicate that dummy motifs receive less relevance than causal motifs.

## C.3  GRAMMAR SATISFIABILITY SCORE (GSS)

GSS treats each grammar as a Boolean specification over attribution-derived predicates. Given a sequence $x$ and signed attributions $a$, we first enumerate motif instances (Appendix D) and compute (i) motif presence predicates (e.g., $\text{HAS}_A(x)$), (ii) mean-sign predicates over motif instances (e.g., $\text{POS}_A(a)$, $\text{NEG}_B(a)$), and (iii) window-based predicates that capture evidence at expected partner locations under spacing constraints (e.g., maximum or minimum mean attribution over 3–5 bp-offset windows).

Each dataset defines two formulas, one for positives and one for negatives, written over these predicates. For a given test example, GSS evaluates the formula corresponding to its true label; the reported score is the fraction of test examples that satisfy the relevant formula. This construction is intended to complement localization metrics by explicitly checking whether attribution maps reflect the rule structure used to generate the label.

## D    APPENDIX D

This appendix summarizes the logical predicates and per-grammar clauses used to compute the Grammar Satisfiability Score (GSS). All predicates are defined per sequence $x$ and its signed attribution map $a \in \mathbb{R}^L$, as implemented in our code.

### D.1    BASIC MOTIF PREDICATES

We work with three fixed motifs:

- MOTIF-A: TATAAA
- MOTIF-B: CCAAT
- MOTIF-D: CGCCAT (dummy motif in **DUMMY**)

For each test sequence $x$ of length $L$, we scan for all non-overlapping instances of MOTIF-A, MOTIF-B, and MOTIF-D. Let $\mathcal{A}(x)$ denote the set of A instances, where each instance $s \in \mathcal{A}(x)$ is a contiguous interval $[\mathrm{start}(s), \mathrm{end}(s)) \subseteq \{1, \dots, L\}$. We define $\mathcal{B}(x)$ and $\mathcal{D}(x)$ analogously for B and D. We compute $\mathrm{mean}(s)$ as the mean attribution between $\mathrm{start}(s)$ and $\mathrm{end}(s)$. We then define simple presence and sign predicates:

$$\mathrm{HAS}_A(x) := |\mathcal{A}(x)| > 0, \qquad \mathrm{HAS}_B(x) := |\mathcal{B}(x)| > 0, \qquad \mathrm{HAS}_D(x) := |\mathcal{D}(x)| > 0$$

$$\mathrm{POS}_A(a) := \frac{1}{|\mathcal{A}(x)|} \sum_{s \in \mathcal{A}(x)} \mathrm{mean}(s) > 0, \qquad \mathrm{NEG}_A(a) := \forall s \in \mathcal{A}(x) : \mathrm{mean}(s) < 0$$

$$\mathrm{POS}_B(a) := \frac{1}{|\mathcal{B}(x)|} \sum_{s \in \mathcal{B}(x)} \mathrm{mean}(s) > 0, \qquad \mathrm{NEG}_B(a) := \forall s \in \mathcal{B}(x) : \mathrm{mean}(s) < 0$$

We also compute a background median by masking out all motif and dummy bases

$$\mathrm{BG\_MEDIAN}(a) := \mathrm{median}\{a_i \mid i \text{ is not part of any A, B, or D instance}\}.$$

We then use

$$\mathrm{BG\_POS}(a) := \mathrm{BG\_MEDIAN}(a) > 0, \qquad \mathrm{BG\_NEG}(a) := \mathrm{BG\_MEDIAN}(a) < 0.$$

### D.2    PARTNER WINDOWS AND SPACING

For grammars with spacing constraints, we define "expected partner" windows that capture where a partner motif would appear if the grammar were satisfied.

**Downstream B windows given A.**    For each A instance $s_A \in \mathcal{A}(x)$, we consider all candidate regions where B could appear 3–5 bp downstream, using the same geometry as the generator:

$$\mathcal{R}_{B|A}(s_A) := \bigcup_{g=3}^{5} [\,\mathrm{end}(s_A) + g,\ \mathrm{end}(s_A) + g + \ell_B\,),$$

where $\ell_B$ is the length of MOTIF-B. For each candidate window, we compute the mean attribution over that window, and aggregate these values either by a maximum (when we expect positive evidence) or a minimum (when we expect negative evidence).

We define:

$$\mathrm{DOWN}^+_{A \to B}(a) := \max_{s_A \in \mathcal{A}(x)} \max_{(u,v) \in \mathcal{R}_{B|A}(s_A)} \frac{1}{v - u} \sum_{i=u}^{v-1} a_i > 0,$$

$$\mathrm{DOWN}^-_{A \to B}(a) := \min_{s_A \in \mathcal{A}(x)} \min_{(u,v) \in \mathcal{R}_{B|A}(s_A)} \frac{1}{v - u} \sum_{i=u}^{v-1} a_i < 0.$$

**Upstream A windows given B.** Symmetrically, for each B instance $s_B \in \mathcal{B}(x)$ we define upstream A windows with a 3–5 bp spacer and derive $\mathrm{UP}^+_{B \to A}(a)$ and $\mathrm{UP}^-_{B \to A}(a)$ in the same fashion.

**Spacing predicate.** We also use a pure location-based spacing predicate:

$$\mathrm{ADJ}_{A,B}(x) := \exists s_A \in \mathcal{A}(x),\ \exists s_B \in \mathcal{B}(x) \text{ such that } 3 \leq \mathrm{start}(s_B) - \mathrm{end}(s_A) \leq 5,$$

### D.3 Grammar-specific satisfiability formulas

For each dataset, we define ground-truth formulas $\mathrm{GSS}_{\mathrm{pos}}$ and $\mathrm{GSS}_{\mathrm{neg}}$ over these predicates. Given a sequence $x$ and attribution map $a$, we evaluate the predicates using $(x, a)$ and evaluate $\mathrm{GSS}_{\mathrm{pos}}$ or $\mathrm{GSS}_{\mathrm{neg}}$ depending on the true label. The GSS reported in the main text is simply the fraction of test sequences where $\mathrm{GSS}_{\mathrm{pos}}$ or $\mathrm{GSS}_{\mathrm{neg}}$ evaluates to $\mathtt{true}$. Below we summarize the intended behavior for each grammar.

**NOT.** Positives have no A; negatives contain A.

- Positive:
$$\mathrm{GSS}^{\mathrm{NOT}}_{\mathrm{pos}}(x, a) := \neg\mathrm{HAS}_A(x).$$

- Negative: A is present and consistently negative. Background is overall positive.
$$\mathrm{GSS}^{\mathrm{NOT}}_{\mathrm{neg}}(x, a) := \mathrm{HAS}_A(x) \wedge \mathrm{NEG}_A(a) \wedge \mathrm{BG\_POS}(a).$$

**OR.** Positives contain at least one of A or B; negatives contain neither.

- Positive: A or B is present and has each have positive mean attribution
$$\mathrm{GSS}^{\mathrm{OR}}_{\mathrm{pos}}(x, a) := (\mathrm{HAS}_A(x) \vee \mathrm{HAS}_B(x)) \wedge \bigwedge_{s \in \mathcal{A}(x) \cup \mathcal{B}(x)} \mathrm{mean}(s) > 0.$$

- Negative:
$$\mathrm{GSS}^{\mathrm{OR}}_{\mathrm{neg}}(x, a) := \neg\mathrm{HAS}_A(x) \wedge \neg\mathrm{HAS}_B(x).$$

**AND_NAND and AND_XOR.** Positives always require both motifs with correct spacing; negatives include all other configurations (only A, only B, both but wrong spacing, or neither).

- Positive:
$$\mathrm{GSS}^{\mathrm{AND}}_{\mathrm{pos}}(x, a) := \mathrm{HAS}_A(x) \wedge \mathrm{HAS}_B(x) \wedge \mathrm{ADJ}_{A,B}(x) \wedge \mathrm{POS}_A(a) \wedge \mathrm{POS}_B(a).$$

- Negative:
  - Case 1: A only (no B). A is necessary for the positive class, so its attribution should remain positive, while the expected B window (3–5 bp downstream) should carry negative evidence:
  $$\mathrm{GSS}^{\mathrm{AND}}_{\mathrm{neg,Aonly}}(x, a) := \mathrm{HAS}_A(x) \wedge \neg\mathrm{HAS}_B(x) \wedge \mathrm{POS}_A(a) \wedge \mathrm{DOWN}^-_{A \to B}(a),$$

  - Case 2: B only (no A) is symmetric:
  $$\mathrm{GSS}^{\mathrm{AND}}_{\mathrm{neg,Bonly}}(x, a) := \mathrm{HAS}_B(x) \wedge \neg\mathrm{HAS}_A(x) \wedge \mathrm{POS}_B(a) \wedge \mathrm{UP}^-_{B \to A}(a).$$

  - Case 3: Neither A nor B adjacent:
  $$\mathrm{GSS}^{\mathrm{AND}}_{\mathrm{neg,none}}(x, a) := \neg\mathrm{HAS}_A(x) \wedge \neg\mathrm{HAS}_B(x),$$

**DUMMY.** Uses the same grammar as **AND_NAND** with an extra constraint to ensure that dummy motifs receive less absolute attribution than causal motifs when present:

- When A or B is present, the absolute attribution on D must be lower than the mean absolute attribution on the causal motifs.
- When neither A nor B is present, the absolute attribution on D must be close to zero (below $10^{-3}$ in our implementation).

**XOR_XNOR.** Positives follow an exclusive-or grammar (exactly one of A or B are present adjacent to each other); Negatives to its complement (both or neither).

- Positive:
  - Case 1: A-only (no B), A is present but should not be evidence for the positive class. Evidence is assigned to the best possible B window

$$\text{GSS}^{\text{XOR}}_{\text{pos,Aonly}}(x, a) := \text{HAS}_A(x) \wedge \neg\text{HAS}_B(x) \wedge \text{NEG}_A(a) \wedge \text{DOWN}^+_{A \to B}(a).$$

  - Case 2: B-only (no A) is symmetric

$$\text{GSS}^{\text{XOR}}_{\text{pos,Bonly}}(x, a) := \text{HAS}_B(x) \wedge \neg\text{HAS}_A(x) \wedge \text{NEG}_B(a) \wedge \text{UP}^+_{B \to A}(a).$$

- Negative:
  - Case 1: Neither A nor B is present

$$\text{GSS}^{\text{XNOR}}_{\text{neg,none}}(x, a) := \left(\neg\text{HAS}_A(x) \wedge \neg\text{HAS}_B(x)\right).$$

  - Case 2: Both A and B adjacent

$$\text{GSS}^{\text{XNOR}}_{\text{neg,both}}(x, a) := \left(\text{HAS}_A(x) \wedge \text{HAS}_B(x) \wedge \text{NEG}_A(a) \wedge \text{NEG}_B(a)\right).$$

**COUNT-3.** Positives have at least three occurrences of A; negatives have fewer than three. A carries positive evidence and the background carries negative evidence:

- Positive:

$$\text{GSS}^{\text{COUNT3}}_{\text{pos}}(x, a) := (|\mathcal{A}(x)| \geq 3) \wedge \text{POS}_A(a) \wedge \text{BG\_NEG}(a).$$

- Negative:

$$\text{GSS}^{\text{COUNT3}}_{\text{neg}}(x, a) := (|\mathcal{A}(x)| < 3) \wedge \text{POS}_A(a) \wedge \text{BG\_NEG}(a).$$

**NIMPLY.** Grammar follows "A and not B" grammar. Positives contain A but not B; negatives cover all other configurations.

- Positive: A is present and positively weighted, the expected B window carries positive evidence, and B itself is absent

$$\text{GSS}^{\text{NIMPLY}}_{\text{pos}}(x, a) := \text{HAS}_A(x) \wedge \neg\text{HAS}_B(x) \wedge \text{POS}_A(a) \wedge \text{DOWN}^+_{A \to B}(a).$$

- Negative:
  - Case 1: Neither A or B adjacent

$$\text{GSS}^{\text{NIMPLY}}_{\text{neg,none}}(x, a) := \neg\text{HAS}_A(x) \wedge \neg\text{HAS}_B(x).$$

  - Case 2: Both A and B present. A remains positively weighted, but B is negatively weighted

$$\text{GSS}^{\text{NIMPLY}}_{\text{neg,both}}(x, a) := \text{HAS}_A(x) \wedge \text{HAS}_B(x) \wedge \text{POS}_A(a) \wedge \text{DOWN}^-_{A \to B}(a) \wedge \text{NEG}_B(a)$$

  - Case 3: B-only: B instance receives negative attribution:

$$\text{GSS}^{\text{NIMPLY}}_{\text{neg,Bonly}}(x, a) := \neg\text{HAS}_A(x) \wedge \text{HAS}_B(x) \wedge \text{NEG}_B(a)$$

  - Case 4: A-only: A is positive; background is overall negative.

$$\text{GSS}^{\text{NIMPLY}}_{\text{neg,Aonly}}(x, a) := \text{HAS}_A(x) \wedge \text{POS}_A(a) \wedge \text{BG\_NEG}(a)$$

## E  LLM USAGE

Claude and ChatGPT were used to assist with manuscript editing. All generated content was reviewed by the authors, who take full responsibility for the final manuscript.

