# OpenReview forum: "Beyond Motif Localization: Probing Rule-Level Signals in Synthetic Genomic Grammars"
_ICLR.cc/2026/Workshop/LMRL — ICLR 2026 Workshop LMRL Poster_

### Official Review · Reviewer_GYM3 · 2026-02-14
**Good Study**

**Rating:** 7
**Confidence:** 4

**Review:**

This paper is well written, clearly structured, and easy to follow. The authors systematically evaluate different attribution methods for both motif localization and rule-level consistency in synthetic genomic datasets. The interpretability methods used are standard and widely adopted, and the evaluation pipeline is carefully designed.

While testing genomic CNN representations using attribution methods is not entirely new, prior work has largely focused on localization alone. This study goes beyond simple motif detection by explicitly evaluating compositional structure and interaction rules. The contribution is therefore incremental but meaningful. Overall, this is a solid workshop submission.

The work is significant because it helps clarify whether genomic deep learning models learn logical and localization rules in synthetic datasets containing motifs. This is important for interpretability and model trust in regulatory genomics.

Strengths of the paper include extensive experiments across multiple attribution methods, the use of various evaluation metrics, and a clear experimental setup. The manuscript is logically organized and easy to read.

There are, however, several limitations. First, only a model with three convolutional blocks is evaluated, as acknowledged by the authors. It would be useful to know whether deeper CNNs, recurrent architectures such as LSTMs or GRUs, or hybrid CNN–RNN models yield different attribution behaviors. Even if this is outside the scope of the current paper, it would be valuable future work.

Second, the study relies entirely on synthetic data. While this allows for controlled evaluation, it may limit biological realism.

Third, the manuscript refers to CGCCAT as “frequent non-causal motif (DUMMY)”. It would be helpful to provide a reference supporting the claim that this motif is non-causal.

Fourth, the choice of 3–5 bp spacing between MOTIF-A and MOTIF-B is unclear. For canonical promoter motifs such as the TATA box and the CCAAT box, spacing is typically much larger. The TATA box is usually located around −25 to −30 relative to the transcription start site, while the CCAAT box most frequently occurs around 80 bp upstream but can range between −50 and −200 (Nussinov et al., 1992). The authors should clarify the rationale for using 3–5 bp spacing in the synthetic rule.

Finally, for the token-level localization AUPRC metric across sequences, it would improve rigor to specify whether the results are macro- or micro-averaged and to clarify how attribution scores were binarized (i.e., what decision threshold was used).

Overall, this is a strong workshop paper with a well-designed evaluation framework and clear experiments. Although limited to a single architecture and synthetic data, it provides useful insights into how attribution methods capture compositional regulatory logic in genomic CNNs.

Reference:
Nussinov R. The eukaryotic CCAAT and TATA boxes, DNA spacer flexibility and looping. J Theor Biol. 1992 Mar 21;155(2):243–70. doi: 10.1016/s0022-5193(05)80597-1. PMID: 1453699.

---

### Official Review · Reviewer_DkwX · 2026-02-23
**limitations of attribution maps in simulated regulatory sequences**

**Rating:** 6
**Confidence:** 4

**Review:**

The authors use simulated regulatory sequences to test whether attribution maps are actually pointing to the important signals.

THis is an important problem because these kinds of maps are widely used as support for deep learning models on regulatory sequences I like the approach of using simulations because this is a great case where we "know" the answer. However, I think the paper seems a bit preliminary/superficial. A few thoughts:

-Majdandzic  et al. 2023 (PMID: 37161475) have already shown that there are issues with standard gradient-based approaches when applied to DNA sequences. It's not clear if the authors are correcting the gradients for these effects. This seems like essential context for the work presented here.

-why not use a simulation approach that creates sequences based on actual biophysical mechanisms rather than simple logic gates? This could be made most convincing done using an evolutionary simulator (e.g., from Saurubh Sinha's group or others). Since the point is to see if models can identify causal mechanisms, logic gates are still relatively abstract, whereas biophysics is closer to what's going on.

---

### Official Review · Reviewer_w3WL · 2026-02-24
**-**

**Rating:** 4
**Confidence:** 3

**Review:**

Summary:

The paper investigates whether attribution methods used in genomics capture compositional regulatory rules, such as motif ordering and spacing, rather than just localizing individual motifs. The authors train a lightweight 1D CNN on synthetic DNA sequences governed by simple, predefined grammar and evaluate various attribution methods using a newly proposed Grammar Satisfiability Score (GSS). In their conclusion, they claim to find that while the methods used localize motifs effectively, they often fail to reflect the intended interaction logic and are sensitive to frequent non-causal patterns.

Strengths:
- I find defining the GSS metric a clever and formal way to evaluate signed attributions and move beyond simple spatial overlap metrics.
- For a tiny paper, the experimental setup is quite comprehensive, accounting for different baseline methods and datasets in the evaluation process.
- The use of cascading randomization to decouple spatial localization from signed structure provides a clear insight regarding the behavior of the models.

Weaknesses:
- The submission is missing an abstract in the actual paper, jumping into the introduction from the get-go – which seems to be similar to the abstract posted on the OpenReview submission but not quite the same.
- Personally, I felt that the writing along the paper is dense and unclear, struggling to clearly motivate the problem to the “average” reader. combined with the inconsistent mentioning of previous work and unstandardized framing, it was difficult for me as a reviewer (and a reader) to catch the paper's core contributions and scope in a single pass.
- While using a lightweight 1D CNN and synthetic data is perfectly acceptable for a "work-in-progress" tiny paper, the authors fail to clearly bridge the gap for the reader regarding how insights from this highly controlled setup can inform the evaluation of more realistic biological models.

Overall, while the proposed GSS metric is conceptually interesting and relevant to the workshop's call for new benchmarks, and the evaluation process was quite thorough, I believe that the paper suffers from significant clarity and formatting issues (including the missing abstract), making the authors' core intentions and the broader significance of the work difficult to understand in its current state.

---

### Meta-Review · Area_Chair_sdFU · 2026-02-28

**Recommendation:** Accept (Poster)
**Confidence:** 4

**Metareview:**

The reviewers raised a number of corrections that are worth addressing, but I agree that this is an interesting direction that's worth discussing at LMLR

---

### Decision · Program_Chairs · 2026-03-02

**Decision:**

Accept (Poster)

**Comment:**

Please see the meta-review.